# Prevention and Killing Efficacy of Carbapenem Resistant *Enterobacteriaceae* (CRE) and Vancomycin Resistant *Enterococci* (VRE) Biofilms by Antibiotic-Loaded Calcium Sulfate Beads

**DOI:** 10.3390/ma13153258

**Published:** 2020-07-22

**Authors:** Paul Stoodley, Jacob Brooks, Casey W. Peters, Nan Jiang, Craig P. Delury, Phillip A. Laycock, Sean S. Aiken, Devendra H. Dusane

**Affiliations:** 1Department of Microbial Infection and Immunity, The Ohio State University, Columbus, OH 43210, USA; Paul.Stoodley@osumc.edu (P.S.); jacob.brooks@osumc.edu (J.B.); caseywilliampeters@gmail.com (C.W.P.); nian.jaing@osumc.edu (N.J.); 2National Centre for Advanced Tribology, Faculty of Engineering and Institute for Life Sciences, University of Southampton, Southampton SO17 1BJ, UK; 3Department of Orthopaedic, The Ohio State University, Columbus, OH 43210, USA; 4Biocomposites Ltd., Keele Science Park, Keele, Staffordshire ST5 5NL, UK; cpd@biocomposites.com (C.P.D.); pl@biocomposites.com (P.A.L.); sa@biocomposites.com (S.S.A.); 5Center for Clinical and Translational Research, The Research Institute at Nationwide Children′s Hospital, 700 Children′s Drive, Columbus, OH 43205, USA

**Keywords:** antibiotic-loaded calcium sulfate beads, biofilm, carbapenem resistant *Enterobacteriaceae*, vancomycin resistant *Enterococci*

## Abstract

Carbapenem-resistant *Enterobacteriaceae* (CRE) and vancomycin-resistant *Enterococci* (VRE) have emerged as multidrug-resistant (MDR) pathogens associated with periprosthetic joint infections (PJI). In this study, we evaluated the efficacy of antibiotic-loaded calcium sulfate beads (ALCSB) in inhibiting bacterial growth, encouraging biofilm formation and killing preformed biofilms of CRE and VRE. Three strains of *Klebsiella pneumoniae* (KP) and a strain of *Enterococcus faecalis* (EF) were used. ALCSB of 4.8-mm diameter were loaded with vancomycin (V) and gentamicin (G), V and rifampicin (R), V and tobramycin (T) or R and meropenem (M), and placed onto tryptic soy agar (TSA), spread with one of the test strains and incubated for 24 h at 37 °C. Beads were transferred daily onto fresh TSA spread plates and the zone of inhibition (ZOI) was recorded until no inhibition was observed. ALCSB containing R + M or R + V produced the most extensive ZOI up to 5 weeks. Biofilm prevention efficacy was investigated by challenging ALCSB daily with 5 × 10^5^ CFU/mL bacterial cells and analyzing for biofilm formation at challenges 1, 2 and 3. In the biofilm killing experiments, ALCSB were added to pre-grown 3-day biofilms of KP and EF strains, which were then analyzed at days 1 and 3 post-exposure. The CFU counts and confocal images of the attached cells showed that ALCSB treatment reduced colonization and biofilm formation significantly (5–7 logs) with combinations of R + M or R + V, compared to unloaded beads. This study provides evidence that the local release of antibiotics from ALCSB may be useful in treating the biofilms of multidrug-resistant strains of CRE and VRE.

## 1. Introduction

In recent years, multidrug-resistant organisms, especially vancomycin-resistant enterococcus (VRE) and carbapenem-resistant enterobacteriaceae (CRE) strains have become more prevalent worldwide, and represent a serious public health threat since they are becoming increasingly resistant to current treatment modalities [1]. In the United States alone, antibiotic-resistant *K*. *pneumoniae* and *Enterococcus* spp. account for nearly 10% of all hospital-acquired infections [2]. Vancomycin resistance among enterococci has markedly increased since it was first described in the mid-1980s [3]. The increasing prevalence of carbapenem-resistant *Enterobacteriaceae*, primarily *K*. *pneumoniae* (KP), is rendering treatment of these infections challenging [4]. Broad-spectrum antibiotic exposure, immune suppression and intravascular devices increase the risk of colonization and infection with one or more antibiotic-resistant bacteria [5].

Periprosthetic joint infection (PJI) is a leading complication and cause of morbidity in joint replacement surgery [6,7]. An underlying problem with treating PJI is that many of the causative pathogens form bacterial biofilms [8]. Whilst Staphylococci are the most common PJI pathogens, VRE and carbapenem-resistant *Klebsiella pneumoniae* strains (a CRE strain) have also been reported, and are associated with poor outcomes [9,10]. Biofilms are microbial populations that adhere to surfaces forming a community inherently highly tolerant to antibiotics compared to their planktonic counterparts, even if isolates are shown to be sensitive by routine clinical microbiological methods, such as the minimum inhibitory concentration (MIC) assay [11]. Biofilms associated with orthopedic infections are typically difficult to diagnose and treat [12,13], and often the only effective intervention is extensive debridement or prostheses removal [14]. Treatment strategies for PJI in total hip and total knee arthroplasty patients include the use of bone cement [14,15], and absorbable calcium sulfate void fillers loaded with antibiotics [16,17], to increase the local concentration and exposure period at the surgical site, beyond those obtainable with systemic administration [18]. In vitro studies show how antibiotic-loaded calcium sulfate beads (ALCSB) can significantly reduce or even eradicate bacterial biofilms of various Gram-positive and -negative PJI pathogens [11,19,20]. ALCSBs are used clinically in total joint replacements. Studies have highlighted the application of ALCSBs for infection prophylaxis and treatment in total joint replacements [21,22,23]. Combinations of two or more antibiotics are commonly used, in local antibiotic delivery in calcium sulfate beads, for patients with hip and knee arthroplasty [17,24,25]. In another study, McPherson et al. (2013) have shown the use of ALCSBs in the treatment of PJI and revision arthroplasty [16]. However, there is little information concerning whether carbapenem-resistant *Klebsiella pneumoniae* and vancomycin-resistant *E. faecalis* strains form biofilms, or how they might be killed by ALCSB. Synergistic interactions between antibiotics have been shown in the treatment of biofilms [26,27]. In a recent study, dose-dependent synergistic interactions of colistin with rifampin, meropenem and tigecycline have been shown to be effective against *K. pneumoniae* biofilms [26]. Powdered rifampicin and vancomycin have shown efficacy against *Staphylococcus aureus* on orthopedic implants [28]. A combination of meropenem and rifampicin has been reported for the treatment of osteomyelitis [29]. We were interested to test the combination of antibiotics loaded in bone cement for use against the multidrug-resistant strains of CRE and VRE. The goals of the present study were as follows: (i) to assess biofilm formation in three CRE strains (including the metallo-beta-lactamase New Delhi strain) and one VRE strain, and (ii) to quantify the long-term bioactivities of various combinations of antibiotics released from ALCSBs, in order to determine the ability to prevent biofilm formation or kill pre-established biofilms of these strains in vitro.

## 2. Materials and Methods

### 2.1. Strains and Culture Conditions

The virulent and multidrug-resistant strains of carbapenem-resistant *Enterobacteriaceae* (CRE) [*Klebsiella pneumoniae* ATCC-BAA 1705 (KP-1705), *K. pneumoniae* ATCC-BAA 2524 (KP-2524), *K. pneumoniae* ATCC-BAA 2146 (KP-2146)] and the vancomycin-resistant *Enterococci* (VRE) strain of *Enterococcus faecalis* ATCC 51,299 (EF-51,299) were used in this study. The strains were obtained from the American Type Culture Collection (ATCC, Manassas, VA), USA and streaked onto tryptic soy agar (TSA) and incubated at 37 °C for 24 h. The colonies were grown on TSA plates and then inoculated into tryptic soya broth (TSB) and incubated at 37 °C for 24 h.

### 2.2. Determination of Minimum Inhibitory Concentration (MIC) of Antibiotics Against CRE and VRE

The MIC was determined using the sterile antibiotic loaded E-test strips (BioMerieux, Durham, NC, USA) following the standard procedure [30,31]. The E-test strips for gentamicin (G), rifampicin (R), tobramycin (T) and vancomycin (V) were used. For some of the strains, the values available in the literature for antibiotic MIC were used (Table 1).

### 2.3. Preparation of Antibiotic-Loaded Calcium Sulfate Beads (ALCSB)

ALCSBs were prepared using Stimulan Rapid Cure^®^ (Biocomposites Ltd., Staffordshire, UK) as detailed previously [11]. ALCSBs were prepared as detailed in Table 2. All the antibiotics were obtained from GoldBio (St. Louis, MO, USA). Unloaded CSBs were used as negative controls. A 20-g pharmaceutical-grade calcium sulfate alpha-hemihydrate powder (Stimulan; Biocomposites Ltd., Keele, UK) was mixed with 6 mL of sterile water (unloaded beads). For antibiotic-loaded beads, tobramycin sulfate powder, vancomycin hydrochloride powder, gentamicin sulfate, rifampicin and meropenem, as shown in Table 2, were mixed with 6 mL sterile water. For preparing the beads, the components were mixed for 30 to 60 s to form a smooth paste, which was pressed into 4.8-mm-diameter hemispherical cavities in a flexible mold. The beads were left undisturbed for 30 to 60 min to set. When set, the beads were removed by flexing the mold.

### 2.4. Modified Kirby–Bauer Assay for Assessing the Repeat Zone of Inhibition (ZOI)

A repeat zone of inhibition (ZOI) test, using a modified Kirby–Bauer assay, was performed to determine the release and potency of antibiotics from a 4.8-mm-diameter ALCSB over a period of 38 days. Briefly, bacteria were spread onto TSA plates using 50 µL of an overnight culture. A single bead was placed onto agar plates with sterile forceps after spreading the bacteria and the plates were incubated at 37 °C for 24 h. ZOI were assessed and photographed, and the beads were transferred onto a freshly spread bacteria on TSA plates. This process was repeated each day until the ZOI were lost. The ZOI were calculated using ImageJ (version 1.48, NIH, Kansas, MO, USA), as described previously [11]. Analyses were performed every day until potency was lost, as evidenced by growth right up to the edge of the bead or the beads having disintegrated. Assays were performed in triplicate, and the data was expressed as the mean of 3 data points with standard error bars.

### 2.5. ALCSBs for Preventing Biofilm Formation

Antibiotic-loaded and unloaded beads (10 beads per well) were placed into 6-well MatTek tissue culture plates (MatTek Corporation, Ashland, MA, USA). The numbers of beads were chosen to compromise between higher bead numbers, which may physically inhibit substratum colonization, and too few beads, which would limit clinical relevance [11]. The MatTek plates were inoculated with 4 mL of an overnight culture diluted with fresh TSB to achieve a concentration of approximately 10^6^ cells/mL and incubated on a shaker incubator set at 37 °C with 5% CO_2_ and at 50 rpm. Every 24 h, the spent medium was replaced, and the beads subjected to a fresh bacterial challenge of 4 mL of 10^6^ cells/mL. At challenges 1, 2 and 3 post-inoculation, CFU counts of the surface-attached bacterial populations in the 6-well plates were performed. Briefly, the wells were rinsed twice in PBS to remove the planktonic cells. The surfaces of the wells were scraped using sterile cell scrapers (VWR, Bridgeport, NJ, USA) to remove the attached biofilms into 1 mL PBS. Following vortexing for 20 s to homogenize the biofilm bacteria, serial dilutions were performed in PBS and plated onto TSA plates using the drop plate method [39,40]. Concurrently, at challenges 1 and 3, the fluorescent BacLight Live-Dead stains for microscopy (ThermoFisher Scientific, Waltham, MA, USA) containing SYTO9 and Propidium iodide (PI) were used to microscopically assess the surface-attached biomass in a second set of MatTek plates. Plates were rinsed with PBS and stained (with 2 µL of BacLight Live-Dead stain per mL of PBS) for 20 min. The plates were gently rinsed and analyzed using a confocal laser scanning microscope with 10 × objective (CLSM, Olympus Fluoview FV10i, Pittsburgh, PA, USA). Cells stained green were live and those in red were determined as dead based on the permeability of the dyes and as per manufacturer’s instructions.

### 2.6. ALCSBs Killing Efficacy of Preformed Biofilms

Each well of a 6-well MatTek plate was inoculated with 4 mL of culture (~10^6^ cells/mL) as described above, but in this case, there were initially no beads in the wells. Biofilms were allowed to form for 3 days (37 °C, 5% CO_2_ and 50 rpm) with daily media exchanges of fresh media. On the 3rd day, 10 antibiotic-loaded or unloaded beads per well were added to each well along with fresh TSB. Every 24 h, the medium was replaced and CFUs were enumerated by sacrificing wells at days 1 and 3 as previously described [11]. Concurrently, at days 1 and 3 post-bead addition, the fluorescent BacLight Live-Dead stains containing SYTO9 and PI were used to microscopically assess the surface-attached biomass in a second set of MatTek plates as described above. The confocal images of prevention and killing experiments were analyzed for biofilm biomass and thickness using COMSTAT analysis (version 2.1, ImageJ plugin). 

### 2.7. Statistics

Data were compared using a paired t-test assuming equal variance, and differences were considered significant when the *p* value was ≤ 0.05 (^⁎^), ≤ 0.01 (^⁎⁎^) and ≤ 0.001 (^⁎⁎⁎^), and non-significant when *p* ≥ 0.05 (ns). Multivariate ANOVA (MANOVA, SPSS version 25, IBM, New York, NY, USA) was performed using SPSS statistical data analysis software.

## 3. Results

### 3.1. Modified Kirby–Bauer Assay for Assessing the Repeat Zone of Inhibition (ZOI)

ALCSBs loaded with G + V, R + V, V + T and R + M elicited zones of inhibition (ZOI) against all bacterial strains tested, but with varying durations of potency, which was both antibiotic- and strain-dependent (Figure 1, Table 1). While all combinations exhibited potency against all strains for at least 5 days, generally the ALCSB loaded with R + V exhibited the greatest duration of potency, ranging from 23 to 38 days while the beads were still intact (Figure 1). The ALCSB loaded with R + M showed extended potency against KP-2524, KP-2146 and EF-51299, but not against the KP-1705 strain. For this strain, ALCSB loaded with G + V showed a potency duration of 20 days, similar to R + V, which was 22 days (Figure 1). The combinations of V + T were least effective against all the test strains, as compared to the R + V and R + M antibiotic combinations tested. This could be due to the lower MIC of antibiotics R and M, as compared to T and V, which showed either higher MIC or resistance against some of the CRE and VRE strains that were tested (Table 1).

### 3.2. ALCSBs for Preventing Biofilm Formation

All the antibiotic combinations demonstrated a significant (*p* < 0.05) retardation of biofilm formation after 24 h, showing a log reduction between 5–7 in CFU/cm^2^ compared to the unloaded control beads (Figure 2). KP-2524 was generally more susceptible to retardation of the formation of its biofilm, using all of the four antibiotic combinations after challenges 1, 2 and 3, as compared to other strains. Against KP-1705, significant log reduction (log 6.0–6.7) was achieved with the R + M combination after 1-, 2- and 3-day bacterial challenges, followed by R + V (with 4.6–6.3 log reduction) and V + T (5.7–5.9). G + V was relatively less effective, with 4.1–4.8 log reduction in bacterial count (Figure 2A). Similarly, in the case of KP-2524, R + M, V + T and R + V were equally effective, with a 6.1–7.6 log reduction, and G + V had the lowest (from 4–4.7 log), as compared to other combinations in reducing the biofilm (Figure 2B). All the antibiotic combinations were less effective in reducing the biofilms of KP-2146, with R + M (~2.5 log), R + V (2.8 log), V + T (1.9) and G + V causing only a 0.8 log reduction after the three daily bacterial challenges (Figure 2C). As expected with repeated challenges (which both removed the antibiotic eluted over the preceding 24 h, and added fresh media and a fresh bacterial challenge), the log reduction in biofilm growth diminished over time. However, there was still significant retardation of biofilm growth in the strains, as mentioned above, for KP-2524 and KP-1705. EF-51299 experienced a similar significant reduction in biofilm growth after the three bacterial challenges, showing >6 log reduction with R + M and V + T, a 5.5 log reduction with R + V and a 4.7 log reduction with G + V, compared to the control (Figure 2D).

Confocal microscopy provided direct evidence that all these strains could form substantial biofilms (Figure 3 unloaded bead control). After 24 h the biofilms were similar in appearance and tended to be relatively flat (Figure 3A). However, after three daily bacterial challenges the biofilms had started to show the structures of aggregates of cells (Figure 3B). At this time, KP-1705 and EF-51,299 were more heterogeneous in structure, and KP-2146 and 2524 had filled in, creating a more uniform layer (Figure 3B). With respect to the effect of the ALCSB, there was general agreement with the CFU data. At days 1 and 3, all the antibiotic combinations were effective in reducing the biomass of biofilms as compared to the unloaded beads, with significant reductions in viability (more cells were red) (Figure 3A,B). Furthermore, it is evident that ALCSB loaded with R + M were more effective in inhibiting biofilm formation than other combinations assessed. Multivariate analysis (ANOVA) also showed significant differences between parameters in the prevention groups (Appendix A). Moreover, the biofilm biomass and biofilm thickness were significantly lower in R + M and R + V, compared to the other antibiotic combinations and the control without antibiotic treatment in prevention group (Appendix A).

### 3.3. ALCSBs Killing Efficacy of Preformed Biofilms

The killing efficacy of pre-established, 3-day biofilms showed a similar trend as was seen with the biofilm prevention assay (Figure 4). After 24 h exposure to the various ALCSB combinations, biofilms were reduced significantly (*p* < 0.05), by between 3 and 7 logs depending on the strain and antibiotic combination. Generally, ALCSB loaded with R + M exhibited the most efficacy, particularly at the longer exposure times, reducing the biofilm bacteria by ~5–6 logs in all strains. KP-1705 showed significant reduction in 3-day biofilms after treatment with R + M (6 log reduction), V + T (6.9 log reduction), G + V (6.7 log reduction) and R + V (5.7 log reduction) (Figure 4A). R + M and R + V were similar in reducing (~5.5 log) the biofilms of KP-2524, followed by G + V and V + T with 3.3 log reduction (*p* < 0.01, Figure 4B). Against KP-2146, R + M was more effective in reducing the biofilms up to 6 logs (*p* < 0.01), followed by R + V (4.7 log), G + V (4 log) and V + T (~3 log) (Figure 4C). R + M and R + V were effective against the VRE strain of EF-51299, with ~5 log reduction (*p* < 0.01), and V + T resulted in a 4.2 log reduction and G + V in a 3.8 log reduction. Multivariate analysis (ANOVA) showed significant differences between different parameters in the killing groups (Appendix A). The reductions in biofilms of EF-51,299 were, however, non-significant (*p* > 0.05) at day 1 of the treatment (Figure 4D). The biofilm biomass and biofilm thickness in the killing group were significantly lower in the R + M and R + V treatment groups, as compared to the other antibiotic combinations and the control without antibiotic treatment (Appendix A).

Confocal microscopy corroborated the CFU data, with unloaded preformed biofilms of KP-1705, 2146, 2524 and EF-51,299 showing thicker biofilms, and green staining with SYTO9 depicting viable cells. These biofilms, when treated with R + V and R + M, showed the killing of biofilms after 1 day of treatment. The residual biofilms were red in color, as compared to the unloaded biofilms and when compared with G + V and V + T (Figure 5A). V + T showed more red cells with KP-2146 and KP-2524 as compared to G + V. Similarly, after treatment with R + M and R + V, the significant killing of biofilms of all strains was evident (Figure 5B). V + T and G + V were less effective in reducing the preestablished biofilms, as evident from the confocal images showing large aggregates of biofilms still present and based on the CFU counts. Overall, ALCSB loaded with R + M showed the greatest reduction in biofilm after 1 and 3 days of treatment (Figure 5A,B).

## 4. Discussion

In this study, we evaluated the effect of Stimulan Rapid Cure antibiotic-loaded calcium sulfate beads on biofilms formed from *Klebsiella pneumoniae* (KP) and *Enterococcus faecalis* (EF) CRE and VRE clinical strains in-vitro. Few of these strains are resistant to gentamicin (G), tobramycin (T) and vancomycin (V) (Table 1). Therefore, combinations of antibiotics were used to determine the effect of ALCSB against these multidrug-resistant strains. Moreover, some of these antibiotic combinations were selected based on their clinical applications in the treatment of joint infections [17,24]. The beads were loaded with various combinations of antibiotics, which are commonly incorporated into bone cement and absorbable bone void fillers to treat PJI (Table 2).

Even though the CRE and VRE strains involved in PJI are relatively uncommon, they present a unique challenge in the era of antimicrobial resistance [10,41,42,43]. Until 2015, the antibiotic treatment regimens for these infections included combinations of agents, which was associated with high toxicity rates (aminoglycosides and colistin), suboptimal pharmacokinetics (aminoglycosides, colistin and tigecycline) and known microbiological resistance [41,43]. In addition to antibiotic resistance, biofilm formation in PJI and other infections associated with foreign bodies has an added virulence factor, since it provides both antibiotic tolerance and protection from host defenses. Biofilm formation by clinical CRE and VRE strains is not well characterized. Therefore, in this study we investigated the efficacy of combinations of potential antibiotic candidates, released from ALCSB, in controlling ATCC strains of *K. pneumoniae* CRE and *E. faecalis* VRE biofilms in-vitro.

In the first set of experiments, a Kirby–Bauer-type diffusion test showed that antibiotics loaded into ALCSB were potent against planktonic bacteria for multiple days, and zones of clearing were maintained for up to 38 days, indicating that eluted antibiotics remained at concentrations high enough to kill the planktonic cells throughout this time. The in vitro elution of antibiotics from high purity calcium sulfate beads has been shown previously in liquid medium [19,44,45,46,47]. Previous studies have shown the comparable time efficacies of ALCSB loaded with V + T against common Gram-negative and Gram-positive PJI species [11,48]. However, here we demonstrate that V + T was less effective in terms of potency duration against the CRE and VRE strains, with potency lost between 7 and 11 days. The in vitro release of R + V retained its potency against all the strains for between 24 and 36 days, and R + M was effective against three strains for 34–35 days. Rifampin (R) has been shown previously to have good anti-biofilm efficacy against Gram-positive bacteria, such as *S. aureus* [49], but should be used in combination with other antibiotics because of the fear of the rapid development of resistance [50,51,52].

Further, studies were performed to determine the effectiveness of the ALCSBs in preventing biofilm formation and killing pre-established biofilms of CRE and VRE strains. We found that in the absence of antibiotics all the strains could form robust biofilms within 3 days. Whether biofilm formation confers additional antibiotic tolerance remains to be seen and would require minimum biofilm eradication concentration (MBEC) and minimal biofilm inhibitory concentration (MBIC) assays, which are standard methods designed specifically for mature biofilms [53]. In a study of the efficacy of 10 different antibiotics (cefazolin, clindamycin, vancomycin, rifampin, linezolid, nafcillin, gentamicin, trimethoprim/sulfamethoxazole, doxycycline and daptomycin) in killing pre-established biofilms formed from 18 *S. aureus* clinical PJI isolates (10 methicillin-resistant and 8 methicillin-sensitive), only rifampin, doxycycline and daptomycin had measurable biofilm MIC values across all *S. aureus* isolates tested [18]. Therefore, the R + M and R + V combinations used in the present study may also demonstrate good efficacy against CRE and VRE strains in PJI. In a previous study, we demonstrated that R + V released from Stimulan ALCSB led to significant reductions in *S. epidermidis* ATCC 35,984 biofilms, a very robust biofilm-forming strain [19]. Along similar lines, we observed that the ALCSB antibiotic combinations significantly reduced biofilm formation, and killed preformed biofilms formed from these CRE and VRE strains, although the degree was dependent on both the strain and the antibiotic combination. Generally, the R + M and R + V combinations performed were the most efficacious in terms of log reductions, in agreement with the repeated ZOI data. Previous studies have found that the MBEC is reduced with time of exposure [54], which is consistent with the understanding that for some antibiotics, both concentration and exposure time are important [55]. In our experiments we saw a general loss of efficacy in retardation and killing over time; however, our assays were particularly challenging. For the prevention assay, the beads were challenged daily with approximately 4 × 10^6^ CFU with fresh media, and any antibiotic that had eluted over the previous 24 h was completely removed, allowing time for the biofilm to grow until inhibitory levels were re-established. On the preformed biofilms, increasing the number of beads or the dose associated with the beads could result in a significant reduction in biofilm formation. This could be a part of the follow-on study to determine the influence of bead number and dose on biofilms. In the biofilm killing assay, although there was no daily challenge, nutrients were replenished, again removing previously eluted antibiotics. Nevertheless, we still saw significant log reductions in biofilm formation (up to 6 logs) after 3 days, the exception being KP-2146, which was more resistant to all antibiotic concentrations and exhibited between 1 and 3 log reductions, depending on the antibiotic combination after three challenges. Similarly, with the biofilm killing assays, there was a 3 to 6 log killing at 3 days with the R + M combination against all strains.

In this series of experiments, R + M and R + V generally performed the best in controlling biofilms of CRE and VRE strains in our in vitro assays. V + T in the US and V + G in Europe are more commonly used to provide coverage against Staphylococci spp. (the most common pathogens in PJI) and broad-spectrum coverage against Gram-negative bacteria, which might be present [56,57,58,59]. One of the issues with treating PJIs is that antibiotic selection is confounded by the difficulty in culturing biofilm bacteria, and often antibiotics administered systemically and locally in cement or in-absorbable mineral beads are often less effective [59,60]. Until more rapid and sensitive methods for isolating bacterial species and generating comprehensive antibiograms are developed, other antibiotic combinations may be considered if CRE or VRE are suspected.

In conclusion, we have shown that strains of CRE and VRE species formed robust biofilms after 3 days, and that Stimulan Rapid Cure antibiotic-loaded calcium sulfate beads retained their potency for up to 38 days. The degree of biofilm retardation and the killing of pre-established biofilms was dependent on the strain and the antibiotic combination, but generally all combinations resulted in significant log reductions for up to three challenges. The combinations of antibiotics rifampicin and vancomycin, and rifampicin and meropenem, were most effective in controlling these CRE and VRE biofilms. Whilst these combinations are effective in vitro, more tests in vivo and clinical investigations are needed.

## Figures and Tables

**Figure 1 materials-13-03258-f001:**
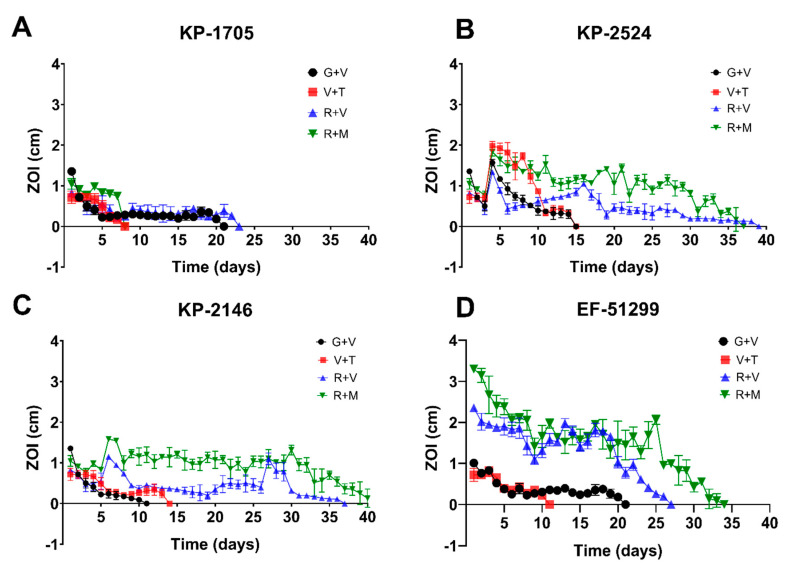
Repeat modified Kirby–Bauer assays for assessing the zones of inhibition (ZOI) of *K. pneumoniae* (**A**) KP-1705, (**B**) KP-2524, (**C**) KP-2146 and (**D**) *E. faecalis* EF-51299 using combinations of antibiotics G + V, R + V, V + T and R + M over a period of 38 days. Data are expressed as means of 3 replicates with standard error bars.

**Figure 2 materials-13-03258-f002:**
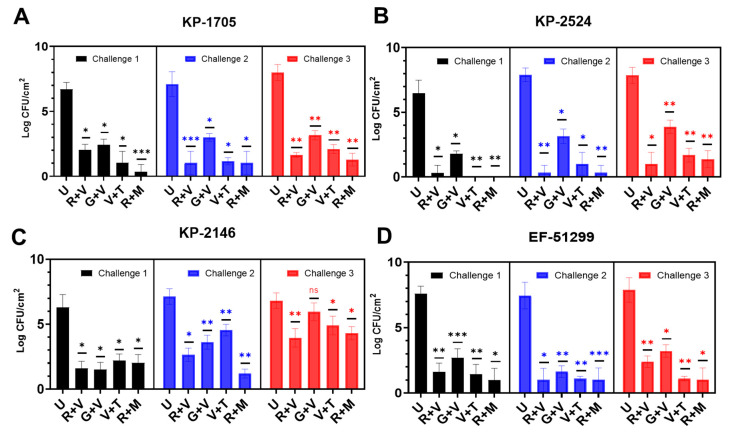
Biofilm formation by (**A**) *K. pneumoniae* KP-1705, (**B**) KP-2524, (**C**) KP-2146 and (**D**) *E. faecalis* EF-51299 in the presence of 10 antibiotic loaded calcium sulfate beads (ALCSBs) as compared to unloaded bead controls (U) as determined by CFU. The differences were significant with *p* values ≤ 0.05 (⁎), ≤ 0.01 (⁎⁎) and ≤ 0.001 (⁎⁎⁎), and non-significant when *p* ≥ 0.05 (ns).

**Figure 3 materials-13-03258-f003:**
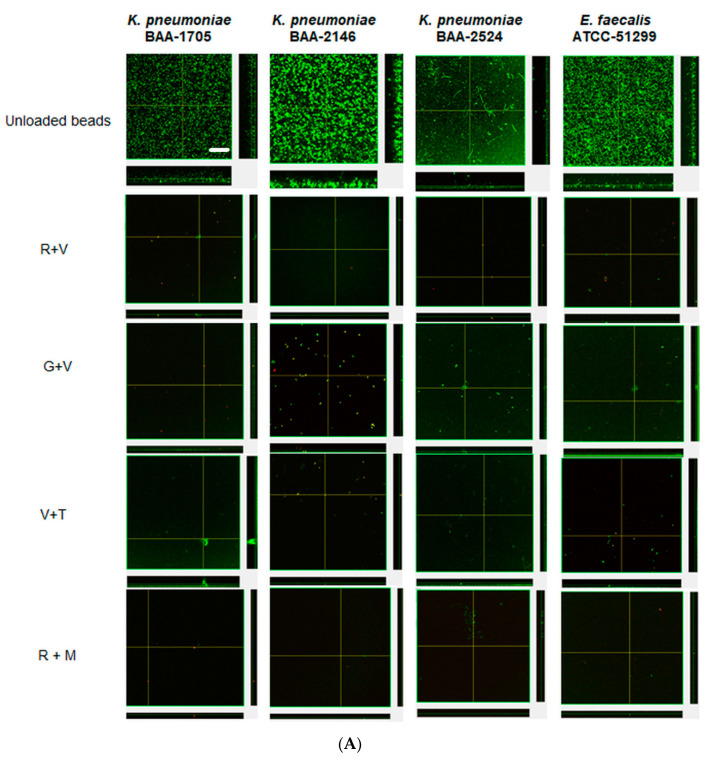
Confocal images of prevention of biofilms at challenge 1 (**A**), challenge 3 (**B**) by ALCSBs, and unloaded control beads. The biofilms were stained with Live-Dead stain where green represents live and red represents dead bacterial cells within biofilms. Bar represents 20 µm size.

**Figure 4 materials-13-03258-f004:**
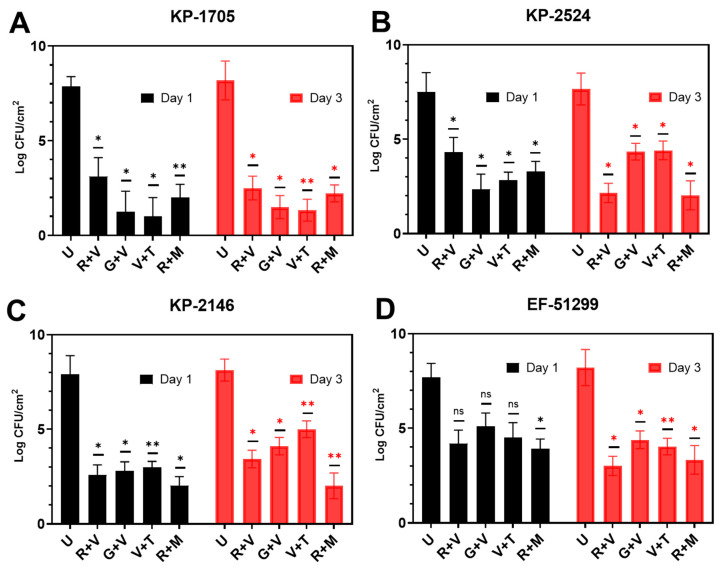
Effect of ALCSB on pre-established biofilms of (**A**) *K. pneumoniae* KP-1705, (**B**) KP-2524, (**C**) KP-2146 and (**D**) *E. faecalis* EF-51299 over time as compared to unloaded bead controls as determined by CFU. The differences were significant with *p* value ≤ 0.05 (⁎), ≤ 0.01 (⁎⁎), and non-significant when *p* ≥ 0.05 (ns).

**Figure 5 materials-13-03258-f005:**
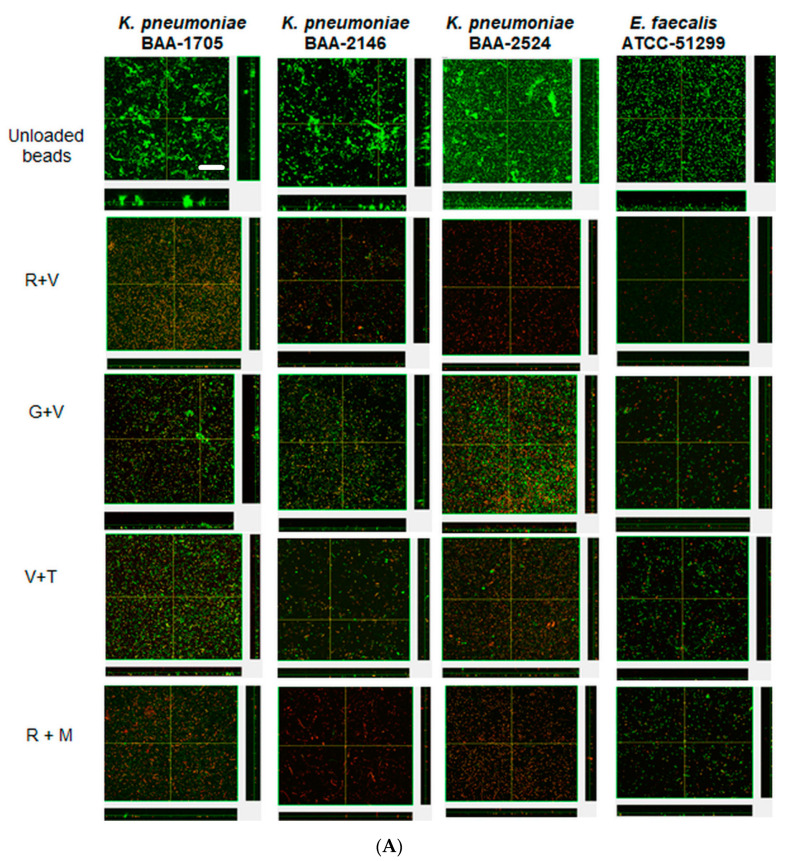
Confocal images showing the effect of ALCSBs against 3-day pre-established biofilms at (**A**) day 1 and (**B**) day 3 post-bead addition. The biofilms were stained with Live-Dead stain, where green represents live and red represents dead bacterial cells within the biofilms. Bar represents 20 µm size.

**Table 1 materials-13-03258-t001:** Minimum inhibitory concentration (MIC) of antibiotics against CRE and VRE strains determined by E-test strip assay; where, KP is *Klebsiella pneumoniae*, EF is *Enterococcus faecalis* and *r* represents resistance.

Antibiotics MIC (µg/mL)	Strains			
	KP-1705	KP-2146	KP-2524	EF-51299
Gentamicin (G)	64	>16 [32]	16	*R*
Rifampicin (R)	64	16 [33]	64	16
Tobramycin (T)	512	>16 [32]	16	*R*
Vancomycin (V)	*r*	>128 [34]	*R*	128
Meropenem (M)	8–64 [35]	16 [36]	43.75 [37]	8 [38]

**Table 2 materials-13-03258-t002:** Antibiotics and amounts used in the study. The amount of antibiotic loaded per pack of Stimulan (10 cc) to prepare beads is mentioned as mg/10 cc, and maximum concentration of antibiotic eluted within 24 h from beads per well during prevention and treatment is mentioned as mg/mL.

Antibiotics	Unloaded (U)	R + V	G + V	V + T	R + M
Rifampicin (R)	0	600 mg/10cc (25 mg/mL)	0	0	600 mg/10 cc (25 mg/mL)
Vancomycin (V)	0	1000 mg/10cc (41.6 mg/mL)	500 mg/10 cc (20.8 mg/mL)	1000 mg/10 cc (41.6 mg/mL)	0
Gentamicin (G)	0	0	240 mg/10 cc (10 mg/mL)	0	0
Meropenem (M)	0	0	0	0	500 mg/10 cc (20.8 mg/mL)
Tobramycin (T)	0	0	0	240 mg/10 cc (10 mg/mL)	0

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
