# Peer review of "Prevention and Killing Efficacy of Carbapenem Resistant Enterobacteriaceae (CRE) and Vancomycin Resistant Enterococci (VRE) Biofilms by Antibiotic-Loaded Calcium Sulfate Beads"

_materials, 2020, doi:10.3390/ma13153258_

Round 1

Reviewer 1 Report

This study investigated and validated the use of the efficacy of antibiotic-loaded calcium sulfate beads 19 for suppression of bacterial growth, biofilm formation and killing of preformed biofilms of Carbapenem-resistant Enterobacteriaceae and vancomycin resistant Enterococci.

In general, the article is written nicely, and the topic looks very interesting.

The language and text flow are perfect.

The Methods and Materials chapter is well written.

The results are presented clearly.

The Discussions section provides a detailed discussion about the current state and results interpretation.

The conclusion part is also well written.

However, the main concern and possibly the main flaw of the study is the Control group. The authors used unloaded beads as a Negative control (vs. beads loaded with antibiotics).

The question: why the authors did not use the same antibiotics (alone) as additional control?

To summarize, I would recommend accepting the manuscript in the current form if the authors can explain and clarify the situation with the control (sham) groups.

Author Response

Reviewer 1:

This study investigated and validated the use of the efficacy of antibiotic-loaded calcium sulfate

beads for suppression of bacterial growth, biofilm formation and killing of preformed biofilms of

Carbapenem-resistant Enterobacteriaceae and vancomycin resistant Enterococci.

In general, the article is written nicely, and the topic looks very interesting.

The language and text flow are perfect.

The Methods and Materials chapter is well written.

The results are presented clearly.

The Discussions section provides a detailed discussion about the current state and results interpretation.

The conclusion part is also well written.

However, the main concern and possibly the main flaw of the study is the Control group. The authors used unloaded beads as a Negative control (vs. beads loaded with antibiotics).

  1. The question: why the authors did not use the same antibiotics (alone) as additional control? To summarize, I would recommend accepting the manuscript in the current form if the authors can explain and clarify the situation with the control (sham) groups.

Response: The combination of antibiotics was used for two purpose:

  1. a) The bacterial strains used in the study are carbapenem resistant strains of Klebsiella pneumoniae and vancomycin resistant strain of Enterococcus faecalis. A few strains are resistant to gentamicin, tobramycin, and vancomycin (Table 1). Therefore, antibiotics alone were not used and instead combination of antibiotics were used to determine the effect of antibiotic loaded calcium sulfate beads (ALCSB) against these multidrug resistant biofilms.
  2. b) The antibiotic combinations were also selected based upon their clinical application in the treatment of joint infections. The combinations are mostly used by surgeons to provide a broad coverage for the treatment of both Gram +ve and Gram -ve bacteria associated with the joint infections. Selected references are included in the revised manuscript that shows the clinical relevance of the antibiotic combinations used in this study.

The following sentence is now included in the discussion section (Page 11, lines 248-252),

‘Few of these strains are resistant to gentamicin (G), tobramycin (T), and vancomycin (V) [Table 1]. Therefore, combinations of antibiotics were used to determine the effect of ALCSB against these multidrug resistant strains.  Moreover, some of these antibiotic combinations were selected based upon their clinical applications in the treatment of joint infections [17,24].’

The following references regarding clinical application of some of the antibiotic combinations loaded in beads are included in the revised manuscript:

  1. Lum, Z.C.; Pereira, G.C. Local bio-absorbable antibiotic delivery in calcium sulfate beads in hip and knee arthroplasty. Journal of orthopaedics 2018, 15, 676-678.

  1. Kallala, R.; Harris, W.E.; Ibrahim, M.; Dipane, M.; McPherson, E. Use of Stimulan absorbable calcium sulphate beads in revision lower limb arthroplasty: Safety profile and complication rates. Bone Joint Res 2018, 7, 570-579, doi:10.1302/2046-3758.710.

Reviewer 2 Report

In the manuscript entitled “Prevention and Killing Efficacy of Carbapenem Resistant Enterobacteriaceae (CRE) and Vancomycin Resistant Enterococci (VRE) Biofilms by Antibiotic-Loaded Calcium Sulfate Beads” the authors evaluated the efficacy of antibiotic-loaded calcium sulfate beads (ALCSB) to inhibit bacterial growth, biofilm formation and killing of preformed biofilms of CRE  and VRE. The results are understandable and well described. I suggest only the following minor changes:

The symbol for litre should be either a uppercase (L) or a lowercase (l), i.e. ml or mL.

Insert space between the number and unit (20 min, 4 mL etc).

In my opinion, the explanation of what the asterisks and “ns” in Figures 2 and 4 mean should be given in Figure captions not only in 2.7 Statistics section.

Unify the reference list. The names of the journals should be given in one format, i.e. full name or abbreviations.

Please, improve resolution of Figure 1.

Author Response

Reviewer 2:

In the manuscript entitled “Prevention and Killing Efficacy of Carbapenem Resistant

Enterobacteriaceae (CRE) and Vancomycin Resistant Enterococci (VRE) Biofilms by Antibiotic

Loaded Calcium Sulfate Beads” the authors evaluated the efficacy of antibiotic-loaded calcium

sulfate beads (ALCSB) to inhibit bacterial growth, biofilm formation and killing of preformed

biofilms of CRE and VRE. The results are understandable and well described. I suggest only the

following minor changes:

  1. The symbol for litre should be either a uppercase (L) or a lowercase (l), i.e. ml or mL.

Response: The symbol for litre has been modified throughout the manuscript and made it uniform as mL.

  1. Insert space between the number and unit (20 min, 4 mL etc.).

Response: The manuscript has been modified and space is inserted between the numbers and units as suggested by the reviewer 2.

  1. In my opinion, the explanation of what the asterisks and “ns” in Figures 2 and 4 mean should be given in Figure captions not only in 2.7 Statisticssection.

Response: The following sentence is now included in Figure 2 and 4 legends (Page 18), ‘The differences were significant with P value ≤0.05 (⁎), ≤0.01 (⁎⁎), ≤0.001 (⁎⁎⁎) and non-significant when P ≥0.05 (ns)’.

  1. Unify the reference list. The names of the journals should be given in one format, i.e. full name, or abbreviations.

Response: The references are now modified, and the format is uniform.

  1. Please improve resolution of Figure 1.

Response: The new figure 1 with color and improved resolution is now included in the revised manuscript.

Reviewer 3 Report

Stoodley et al. submitted an article for publication that deals with the investigation of prevention and killing efficacy of CRE and VRE biofilms by ALCSB. The authors show that local release of antibiotics (using ALCS Beads) reduced CRE and VRE colonization and biofilm formation. Against planktonic bacteria, ALCSB were highly effective. However, when grown as biofilm, the effect is less pronounced. These results may be of interest to the readership of the Materials journal.

However, please expand the discussion particularly about why ALCSB were more effective on bacterial lawn (not planktonic) but not on biofilms (including preformed). 

My comments

Line 97. Table 2 are the number concentrations in ug/mL? please add this information to table or footnote. 

lines 93-97. A brief description on how beads are prepared should be added.

Line 98. can the author investigate the effect of the beads (10 beads) in liquid culture instead Kirby-Bauer assay (MICs using liquid culture).How much antibiotic is released over time? (How long you will need to reach to killing concentration)?

line 110. can you add the final concentration of ATBs that is associated with 10 beads. 

For preventing biofilm formation:When added to the wells, bacteria are still in planktonic condition and if beads are already present, they will at first release antibiotic at subMIC levels, these sub-lethal doses could trigger biofilm formation. Once formed these biofilms would then resist and tolerate the higher doses of ATBs. Once they added the beads to the well, author should add fresh media at first and let the beads releasing antibiotics and then add bacterial suspension.

line 127. On preformed biofilm, what would happen if you increase the number of beads or the doses associated with beads. Better killing/eradication? In the discussion, authors should expand little about this. 

line 173 EF-51299 instead of EF-5129

lines 180-181. Authors should add more details about thickness of biofilm and biomass volume. This would help to understand how biofilm tolerate ALCSB. 

190. Confocal images. please add on the footnotes what fluorescent was used and represent each color (green=live and red=dead). 

Author Response

Reviewer 3:

Stoodley et al. submitted an article for publication that deals with the investigation of prevention

and killing efficacy of CRE and VRE biofilms by ALCSB. The authors show that local release of

antibiotics (using ALCS Beads) reduced CRE and VRE colonization and biofilm formation.

Against planktonic bacteria, ALCSB were highly effective. However, when grown as biofilm,

the effect is less pronounced. These results may be of interest to the readership of the Materials

journal.

  1. However, please expand the discussion particularly about why ALCSB were more effective on bacterial lawn (not planktonic) but not on biofilms (including preformed). 

Response: To avoid confusion, we have changed the M&M section (Page 6, lines 120-131) of Modified Kirby-Bauer assay for assessing the repeat zone of inhibition; wherein, we spread the bacteria and placed the ALCSB and incubated for 24 h. We would like to point out that there were no pre-grown bacterial lawns rather planktonic cells spread on TSA that were subjected to the treatment with ALCSB. After 24 h, the zone of inhibition was measured, and the bead was transferred onto other TSA plate that was spread with the test culture before placing the bead for repeat zone of inhibition. This is in fact ZOI against planktonic cells and not the biofilm lawn. Therefore, the killing (analyzed by ZOI) against planktonic bacteria on agar surface was greater than the biofilms (prevention and killing).

My comments

  1. In Table 2, are the number concentrations in µg/mL? please add this information to table or footnote. 

Response: The concentration of antibiotics used to prepare beads were in mg of antibiotic/ 10cc Stimulan. The maximum amount of antibiotic that will be released from 10 beads during the prevention and killing experiments is now updated in Table 2 (Page 18).

There are studies which shows the elution of independent antibiotics in liquid medium. The references (19, 37-40) are included in the revised manuscript. This information is also updated in the discussion section on page 12, lines 268-269.

  1. Lines 93-97. A brief description on how beads are prepared should be added.

Response: The following description on how beads were prepared is now included in the M&M section (Pages 5-6, lines 111-118). A 20 g pharmaceutical-grade calcium sulfate alpha-hemihydrate powder (Stimulan; Biocomposites Ltd., United Kingdom) was mixed with 6 mL of sterile water (unloaded beads). For antibiotic loaded beads, tobramycin sulfate powder, vancomycin hydrochloride powder, gentamicin sulfate, rifampicin, meropenem as shown in Table 2 were mixed with 6 ml sterile water. For preparing the beads, the components were mixed for 30 to 60 s to form a smooth paste, which was pressed into 4.8-mm-diameter hemispherical cavities in a flexible mold. The beads were left undisturbed for 30 to 60 min to set. When set, the beads were removed by flexing the mold.

  1. Line 98. Can the author investigate the effect of the beads (10 beads) in liquid culture instead Kirby-Bauer assay (MICs using liquid culture)? How much antibiotic is released over time? (How long you will need to reach to killing concentration)?

Response: The objective of the zone of inhibition study was to investigate the potency of ALCSB (using Kirby-Bauer assay) over the period of 38 days, and to determine which combination is effective over longer periods. We agree with the reviewer’s suggestion to investigate the elution of beads in liquid culture; however, this could be a part of the follow-on study to determine the elution kinetics and efficacy of different antibiotics loaded in calcium sulfate beads either alone and in combinations.

It should also be noted that unless carefully designed to mimic the conditions with the Kirby-Bauer method, elution data may be of limited informative value, as experimental conditions in elution experiments can have dramatic effects on the outcomes. Care needs be taken when drawing comparisons of antibiotic elution between different studies, as the design of elution experiments can have a significant effect on the levels and duration of antibiotic release.  Within reported literature, there is a large variation in the in-vitro experimental methods used to determine antibiotic elution and methodologies also vary with respect to a number of parameters, such as:

  1. The volume removed for analysis and the eluent sampling intervals: Complete, frequent exchange of fluid will typically result in antibiotic elution for a shorter duration than partial exchange of fluid [1].  These alternative fluid exchange methods are used to model environments with high and low fluid dynamics, respectively.
  2. The quantity of material tested: Increasing the ratio of material to the volume of fluid will usually increase the concentration of antibiotic in the fluid [2].
  3. The nature by which the sample is presented to the solution: Elution from a single small bead [3-5] will typically have lower antibiotic concentrations with elution for a shorter duration than if a larger cast cylinder of material is used [6, 7].
  4. The volume and nature of the solution into which the sample is immersed: Increasing the ratio of the volume of fluid to the quantity of material used will typically lower antibiotic concentrations.  Different fluids will result in different antibiotic concentrations eluted into the fluid.

The elution of antibiotics could therefore be an appropriate follow-on stand-alone study.

References

  1. McLaren, A.C., et al., The effect of sampling method on the elution of tobramycin from calcium sulfate. Clin Orthop Relat Res, 2002(403): p. 54-7.
  2. Aiken, S.S., et al., Local Release of Antibiotics for Surgical Site Infection Management Using High-Purity Calcium Sulfate: An In Vitro Elution Study. Surg Infect (Larchmt), 2014.
  3. Wichelhaus, T.A., et al., Elution characteristics of vancomycin, teicoplanin, gentamicin and clindamycin from calcium sulphate beads. J Antimicrob Chemother, 2001. 48(1): p. 117-9.
  4. Parker, A.C., et al., Evaluation of Two Sources of Calcium Sulfate for a Local Drug Delivery System: A Pilot Study. Clin Orthop Relat Res, 2011.
  5. Miclau, T., L.E. Dahners, and R.W. Lindsey, In vitro pharmacokinetics of antibiotic release from locally implantable materials. J Orthop Res, 1993. 11(5): p. 627-32.
  6. Kanellakopoulou, K., et al., In vitro elution of daptomycin by a synthetic crystallic semihydrate form of calcium sulfate, Stimulan. Antimicrob Agents Chemother, 2009. 53(7): p. 3106-7.
  7. Panagopoulos, P., et al., In vitro elution of moxifloxacin and fusidic acid by a synthetic crystallic semihydrate form of calcium sulphate (Stimulan). Int J Antimicrob Agents, 2008. 32(6): p. 485-7.

  1. line 110. can you add the final concentration of ATBs that is associated with 10 beads? 

Response: The concentration of antibiotics per bead basis is included in Table 2 under the concentration of respective antibiotics. The maximum concentration of antibiotic that will be eluted from the beads in the prevention and treatment assays in the first 24 h period would be the amount of antibiotic per bead x 10 beads / volume of media.

Table 2. Antibiotics and amounts used in the study. The amount of antibiotic loaded per pack of Stimulan (10cc) to prepare beads is mentioned as mg / 10 cc and maximum concentration of antibiotic eluted within 24 h from beads per well during prevention and treatment is mentioned as mg/ mL.

  1. For preventing biofilm formation: When added to the wells, bacteria are still in planktonic condition and if beads are already present, they will at first release antibiotic at subMIC levels, these sub-lethal doses could trigger biofilm formation. Once formed these biofilms would then resist and tolerate the higher doses of ATBs. Once they added the beads to the well, author should add fresh media at first and let the beads releasing antibiotics and then add bacterial suspension.

Response: The reviewer suggestions are useful for future follow-on experiments and care will be taken to first add the beads to the wells of the 6-well plate to allow the antibiotics to release and then add bacterial suspension.

  1. line 127. On preformed biofilm, what would happen if you increase the number of beads or the doses associated with beads. Better killing/eradication? In the discussion, authors should expand little about this. 

Response: The following information was included in the discussion section  (Page 14, lines 299-301): ‘On the preformed biofilms, increasing the number of beads or the dose associated with the beads could result in significant reduction in biofilm formation’.

The number of beads were chosen to compromise between higher bead numbers, which may physically inhibit substratum colonization, and too few beads, which would limit clinical relevance (Page 6, lines 135-137). The bead numbers were selected based on our previously published study (reference 11).

  1. Howlin, R.; Brayford, M.; Webb, J.; Cooper, J.; Aiken, S.; Stoodley, P. Antibiotic-loaded synthetic calcium sulfate beads for prevention of bacterial colonization and biofilm formation in periprosthetic infections. Antimicrobial agents and chemotherapy 2015, 59, 111-120.

  1. Line 173 EF-51299 instead of EF-5129.

Response: The correction has been made as EF-51299 on page 9, lines 200 and 207.

  1. Lines 180-181. Authors should add more details about thickness of biofilm and biomass volume. This would help to understand how biofilm tolerate ALCSB. 

Response: The information about biofilm thickness and biovolume for prevention and killing of three CRE and one VRE strains using ALCSB is now included as the supplementary information (Table. S2) and in the results section (Page 10, lines 212-214 and Page 11, lines 231-233).

  1. Confocal images. please add on the footnotes what fluorescent was used and represent each color (green=live and red=dead). 

Response: The figure legends (figures 3 and 5, Pages 19-20) are now modified to read, ‘The biofilms were stained with Live-Dead stain where green represents live and red represents dead bacterial cells within biofilms.’